# Synthesis of Half-Titanocene Complexes Containing π,π-Stacked Aryloxide Ligands, and Their Use as Catalysts for Ethylene (Co)polymerizations

**DOI:** 10.3390/polym14071427

**Published:** 2022-03-31

**Authors:** Jin Gu, Xiaohua Wang, Wenpeng Zhao, Rui Zhuang, Chunyu Zhang, Xuequan Zhang, Yinghui Cai, Wenbo Yuan, Bo Luan, Bo Dong, Heng Liu

**Affiliations:** 1Shandong Provincial Key Laboratory of Olefin Catalysis and Polymerization, Chambroad Chemical Industry Research Institute Co., Ltd., Qingdao 266042, China; gujin930118@163.com (J.G.); rui.zhuang@chambroad.com (R.Z.); yinghui.cai@chambroad.com (Y.C.); wenbo.yuan@chambroad.com (W.Y.); bo.luan@chambroad.com (B.L.); 2Key Laboratory of Rubber-Plastics, Ministry of Education/Shandong Provincial Key Laboratory of Rubber-Plastics, Qingdao University of Science & Technology, Qingdao 266042, China; bh146@qust.edu.cn (X.W.); zwp@qust.edu.cn (W.Z.); cyzhang@qust.edu.cn (C.Z.); xqzhang@qust.edu.cn (X.Z.); 3Changchun Institute of Applied Chemistry, Chinese Academy of Sciences, Changchun 130022, China

**Keywords:** half-titanocene complexes, π,π-stacking interaction, fused-aryloxide ligands, ethylene (co)polymerization

## Abstract

A family of half-titanocene complexes bearing π,π-stacked aryloxide ligands and their catalytic performances towards ethylene homo-/co- polymerizations were disclosed herein. All the complexes were well characterized, and the intermolecular π,π-stacking interactions could be clearly identified from single crystal X-ray analysis, in which a stronger interaction could be reflected for aryloxides bearing bigger π-systems, e.g., pyrenoxide. Due to the formation of such interactions, these complexes were able to highly catalyze the ethylene homopolymerizations and copolymerization with 1-hexene comonomer, even without any additiveson the aryloxide group, which showed striking contrast to other half-titanocene analogues, implying the positive influence of π,π-stacking interaction in enhancing the catalytic performances of the corresponding catalysts. Moreover, it was found that addition of external pyrene molecules was capable of boosting the catalytic efficiency significantly, due to the formation of a stronger π,π-stacking interaction between the complexes and pyrene molecules.

## 1. Introduction

π,π-stacking refers to the *π*-interaction between the π-electron clouds of aromatic systems [1,2]. It is mainly caused by intermolecular overlapping of p orbitals in π-conjugated systems. Based on its stacking patterns, π,π-stacking can be classified into three models: face-to-face (sandwich), edge-to-face (T-shaped), and offset face-to-face (parallel-displaced) [3,4]. Due to its multiplicity and ubiquity, such a non-covalent interaction has been widely explored in many fields of chemistry [5,6,7,8] and biochemistry [9,10,11], and more importantly, it also reveals a decisive role in influencing the course of a reaction [12,13,14,15,16,17,18,19,20,21]. However, regarding olefin polymerizations, the influence of π,π-stacking on catalytic performances is still much less explored. As the field progressed, the main strategy for regulating olefin polymerization behaviors from a catalyst level is still relying on steric and electronic modification of the ligands, and for a long time, scientists have been seeking for effective alternative methodologies [22,23]. Considering its diversity as well as facile construction from simple introducing fused-aryl moieties, π,π-stacking might act as a promising candidate for realizing such a goal, and in recent years, research interest in this field is upsurging (Figure 1). For instance, incorporation of intra-ligand π,π-interaction is able to improve the thermal robustness of the active species for α-diimine Ni/Pd mediated ethylene (co)polymerizations, and simultaneously regulate the molecular weights and branching densities of the resultant polyethylenes [24,25]; immobilization group IV metallocene and bis(arylimino)pyridine ferrous complexes onto graphene nanoplatelets or carbon nanotubes via *π*,*π*-stacking interactions is capable of enhancing the overall catalytic activities towards olefin polymerization [26,27], and in some cases, affording ultra-high-molecular-weight products, that is difficult to be achieved by traditional catalysts [28,29].

Half-titanocene type complexes Cp’TiX_2_(OR) (Cp’ = substituted cyclopentadienyl; X = Cl, Me etc.; OR = alkyloxide, aryloxide, etc.), is currently one of the most important systems that have been widely explored for ethylene (co)polymerizations [30,31,32,33,34,35,36,37,38,39,40,41,42,43]. In such a system, due to the abundance and commercial availability of diversified phenol derivatives, ethylene (co)polymerization performances as well as the molecular parameters, such as molecular weight, polydispersity, comonomer incorporation percentage, comonomer sequences, etc., can be well regulated through tailoring the substituents on the phenoxide moiety. Based on such considerations, in this research, a series of half-titanocene complexes containing fused-aryloxides ligands were disclosed, and intermolecular π-π stacking interaction can be clearly observed between these aryloxide moieties. Their structural characterizations, as well as the influence of π-π stacking interaction on ethylene homo-/co- polymerization are also studied, which will be given in the following.

## 2. Experimental Section

### 2.1. Materials

All manipulations of air- and moisture-sensitive materials were carried out in a high vacuum line or a glovebox with a medium capacity recirculator (<2 ppm oxygen). The solvents (hexane, toluene, dichloromethane, benzene-*d*^6^) were purchased from Shanghai Aladdin Biochemical Technology Co., Ltd. (Shanghai, China) and refluxed over by sodium or CaH_2_ and degassed by three freeze–pump–thaw cycles prior to use. The Trichloro (cyclopentadienyl) titanium and trichloro (pentamethylcyclopentadienyl) titanium were supplied by Merck Ltd. (Shanghai, China) on Aldrich Chemical Company. DMAO was evaporated under vacuum to obtain a white residue according to the literature [44].

### 2.2. Characterizations

^1^H NMR (400 MHz) and ^13^C NMR (100 MHz) spectra of complexes measured on a Bruker-300 MHz (Bruker Optics, Ettlingen, Germany) in C_6_D_6_ using tetramethylsilane as an internal standard. Ultraviolet−visible (UV−vis) absorption spectra were recorded on a Cary 500 Scan UV−vis spectrophotometer. For the absorption of the UV spectra, the concentration of pyrene was fixed at 5 × 10^−6^ M and the concentration of the host was increased from 0 to 24 × 10^−7^ M in CH_2_Cl_2_ at 298 K. The NMR spectra of the polymers were recorded on a Varian Unity-400 NMR (Varian, Inc., Palo Alto, CA, USA) spectrometer at 135 °C with C_6_D_4_Cl_2_ as a solvent. Elemental analysis was carried out using an elemental Vario EL spectrophotometer (Elementar Analysensysteme GmbH, Langenselbold, Germany). The molecular weights (*M*_n_) and molecular weight distributions (PDI, *M*_w_/*M*_n_) of polymers were determined by PL-GPC 200 high-temperature gel permeation chromatography (Agilent Technologies, CA, USA) at 135 °C using 1,2,4-Trichlorobenzene as an eluent. The melting points of the ethylene/1-hexene copolymers were determined on a TA DSC Q20 instrument (TA, New Castle, DE, USA) at a heating/cooling rate of 10 °C/min. All the DFT calculations were performed with the Gaussian 09 program [45]. The B3LYP functional together with the 6-311+G** basis set for all the atoms. Solvent (toluene) effects were included using the SMD model [46]. The 3D molecular structures displayed in the manuscript were drawn by using CYLview [47].

Crystals of the titanium complexes were obtained by laying hexane onto toluene solutions. Data collections were performed on a Bruker SMART APEX diffractometer at −88.5 °C with a CCD area detector using graphite monochromated MoK radiation (λ = 0.71073 Å). The determination of crystal class and unit cell parameters was carried out by the SMART program package. The raw frame data were processed using SAINT and SADABS to collect the reflection data file. Refinement was performed on F^2^ anistropically for all non-hydrogen atoms by full-matrix least-squares method. Details of X-ray structure determinations and refinements are summarized in Appendix A. CCDC numbers for Ti1 and Ti3: 1874219, 1481991.

### 2.3. Synthesis of Half-Titanocene Complexes

#### 2.3.1. Synthesis of Complex Ti1

A solution of the CpTiCl_3_ (0.5 g, 2.27 mmol) in 10 mL of CH_2_Cl_2_ was reacted with 1.0 *equiv*. of lithium 1-naphthoxide (0.33 g, 2.27 mmol) in 10 mL CH_2_Cl_2_. The mixture was warmed from −78 °C to room temperature and stirred for 12 h. The solvent was evaporated under vacuum to obtain a red residue. The powder was washed twice with diethyl ether (10 mL) and filtered, recrystallization from the concentrated toluene/hexane solution afforded the target complex as red crystals. Yield: 62%. ^1^H NMR (CDCl_3_): δ 8.49–8.47 (m, 1H, Ar-H), 7.57–7.55 (m, 1H, Ar-H), 7.35–7.27 (m, 2H, Ar-H), 7.23–7.19 (m, 1H, Ar-H), 7.09–7.05 (m, 1H, Ar-H), 6.82–6.80 (m, 1H, Ar-H), 6.08 (s, 5H, Cp). ^13^C NMR (126 MHz, CDCl_3_) δ 164.97, 134.48, 127.75, 126.92, 126.70, 125.59, 125.47, 124.66, 122.24, 121.15, 114.66. Anal. Calcd for C_15_H_12_Cl_2_OTi: C, 55.09; H, 3.70. Found: C, 55.29; H, 3.65.

#### 2.3.2. Synthesis of Complex Ti2

The complex Ti2 was carried out using a similar method as preparation of Ti1. Yield: 63%. ^1^H NMR (CDCl_3_): δ 8.59–8.57 (m, 1H, Ar-H), 8.39–8.35 (m, 2H, Ar-H), 7.58–7.33 (m, 5H, Ar-H), 7.20 (s, 1H, Ar-H), 6.07 (s, 5H, Cp). ^13^C NMR (126 MHz, CDCl_3_) δ 163.28, 131.76, 131.18, 128.31, 128.10, 127.68, 127.33, 126.19, 122.97, 122.83, 122.77, 121.24, 119.60, 113.50. Anal. Calcd for C_19_H_14_Cl_2_OTi: C, 60.52; H, 3.74. Found: C, 60.63; H, 3.70.

#### 2.3.3. Synthesis of Complex Ti3

Lithium1-pyrenoxide (0.49 g, 2.27 mmol) was added slowly to a stirred toluene solution (10 mL) containing Cp*TiCl_3_ (0.65 g, 2.27 mmol) at −78 °C. The mixture was warmed to room temperature and then refluxed for 24 h. The red powder was obtained by removing the solvent, recrystallization from the concentrated toluene/hexane solution afforded the desired product as red crystals. Yield: 60%. ^1^H NMR (400 MHz, CDCl_3_, δ, ppm): 8.42–8.40 (m, 1H, Pyrene-H), 8.20–7.85 (m, 7H, Pyrene-H), 7.80–7.71 (m, 1H, Pyrene-H), 6.82 (s, 5H, Cp-H). ^13^C NMR (126 MHz, CDCl_3_) δ 163.45, 131.38, 128.31, 127.13, 126.88, 126.55, 125.64, 125.32, 125.16, 121.17, 120.83, 119.56, 117.21, 103.81. Anal. Calc. for C_21_H_14_Cl_2_OTi (401.1): C, 62.88; H, 3.52. Found: C, 62.91; H, 3.49.

#### 2.3.4. Synthesis of Complex Ti4

The complex Ti4 was carried out using a similar method as preparation of Ti3. Yield: 42%. ^1^H NMR (400 MHz, CDCl_3_, δ, ppm): 8.42–8.40 (m, 1H, Pyrene-H), 8.20–7.85 (m, 7H, Pyrene-H), 7.80–7.71 (m, 1H, Pyrene-H), 1.5 (s, 15H, Cp-Me). ^13^C NMR (126 MHz, CDCl_3_) δ 159.51, 133.24, 131.51, 127.62, 127.28, 126.33, 126.19, 125.57, 125.39, 125.12, 124.84, 121.34, 118.33, 13.13. Anal. Calc. for C_26_H_24_Cl_2_OTi (471.2): C, 66.27; H, 5.13. Found: C, 66.21; H, 4.79.

### 2.4. Polymerization Procedure

A typical polymerization procedure for ethylene polymerization was shown as follows: 100 mL stainless steel autoclave was heated in a vacuum at 80 °C and recharged with ethylene three times, then cooled to room temperature. In a 10 mL Schlenk flask, the additive (pyrene) solution in toluene (1 mL) was added to a solution of titanium complex Ti1, the mixture stirred for 10 min and transferred into the reactor. Then, the required amount of the cocatalyst was added, the autoclave was pressurized to 6 atm immediately. The reaction mixture was stirred at the desired temperature for 10 min. The mixture was then quenched by pouring into a large quantity of acidified ethanol containing HCl (3 M). The polymer was collected by filtration, washed with water and ethanol, and dried to a constant weight under vacuum at 70 °C.

## 3. Results and Discussion

### 3.1. Synthesis and Characterization of the Half-Titanocenes Ti1–Ti4

Half-titanocene complexes **Ti1**–**Ti4** containing anionic fused-aryloxide ligands were prepared by stoichiometric reaction between CpTiCl_3_ (or Cp*TiCl_3_) and newly prepared lithium aryloxide derivatives (Figure 2). Additionally, very pure products could be crystallized as red platelets in high yields upon cooling their saturated n-hexane/toluene solutions to −35 °C in the drybox. In order to establish the structure-activity relationship, fused-aryloxides bearing different π-systems, including 1-naphthoxide, 9-phenanthrenoxide, 1-pyrenoxide, were intentionally explored. All the complexes were well-characterized by NMR and elemental analysis. Moreover, the solid-state structure of **Ti1** and **Ti3** were further confirmed by single crystal X-ray analysis.

Single crystal structures of complexes **Ti1** and **Ti3** are shown in Figure 1, Figure 2, Figure 3 and Figure 4. In these two complexes, the Ti-O and O-C_ipso_ bond distances are 1.7788(15) Å (**Ti1**), 1.7794(18) Å (**Ti3**) and 1.365(2) Å (**Ti1**), 1.362(3) Å (**Ti3**), respectively, which are quite similar to previously reported CpTiCl_2_(OAr) analogues that reveal Ti-O bond distances of 1.75-1.82 Å and O-C_ipso_ bond distances of ca. 1.36 Å [34,42,48,49,50,51,52,53,54,55,56,57,58]. In contrast, they reveal much larger C_ipso_-O-Ti bond angles (158.09(13)^o^ for **Ti1**, 158.26(18)^o^ for **Ti3**) when comparing half-titanocenes ligated with 2,6-unsubstituted aryloxide moieties that possess similar steric hindrance around the metal center, such as C_ipso_-O-Ti bond angle of 153.77(16)^o^ in CpTiCl_2_(O(4-^t^BuPh) [59]. These larger angles imply much bigger O→Ti π donations into titanium due to the much bigger π systems in fused-aryloxide moieties. Nevertheless, they are still comparatively smaller than counterparts having 2,6-diisopropylphenoxide ligand (163.0(4)^o^ for CpTi, and 173.0(3)^o^ for Cp*Ti) due to the lack of ortho- bulky groups that could ‘sterically’ force the more open C_ipso_-O-Ti angle [31].

As designed, intermolecular π-π stacking interactions can be clearly observed in both two complexes (Figure 1, Figure 2, Figure 3 and Figure 4). Two spatially adjacent anionic fused-aryloxide groups are found to be almost parallel with each other, giving a reversely orientated dimer structure. Additionally, similarly to most cases, an offset stacked conformation was adopted [1,4]. The strength of the π-π stacking interactions can be evaluated by the distances between two almost parallel planes. As illustrated in Figure 2 and Figure 4, an obvious shorter distance with value of 3.430 Å in **Ti3** was observed (versus 3.519 Å in **Ti1**), implying the much stronger π-π interaction in **Ti3**. This result makes sense when considering the overlapping nature of p orbitals in π-conjugated systems, which becomes stronger as the number of π-electrons increases.

### 3.2. Ethylene (Co)polymerization Performances

Ethylene homopolymerizations were firstly evaluated by using the present half-titanocene complexes **Ti1**–**Ti4** bearing intermolecular π-π stacking interactions. Dry methylaluminoxane (DMAO), which was prepared by removing free trimethylaluminum from commercially available MAO toluene solution [44], was chosen as cocatalyst herein because it had been previously testified to be effective for achieving high catalytic efficiencies as well as high molecular weight products in analogous half-titanocene mediated olefin polymerizations [33]. As the results summarized in Table 1, **Ti1** and **Ti2**, bearing 1-naphthoxide and 9-phenanthrenoxide moieties, respectively, gave very similar catalytic activities of 4.98 × 10^6^ and 5.07 × 10^6^ g PE•mol^−1^ (Ti)•h^−1^; nevertheless, for **Ti3** promoted systems, much lower polymer yields were afforded under identical conditions. Due to the structure similarities of **Ti1** and **Ti3** in Ti-O and O-C_ipso_ bond distances and C_ipso_-O-Ti bond angles that had been concluded from single crystal data, such catalytic differences in **Ti1**, **Ti2** and **Ti3** were probably originated from steric reasons caused by the π-π stacked dimer structure. As the steric crowding maps from buried volume calculations for complexes **Ti1** and **Ti3** (Figure 5) [60,61,62], **Ti3** revealed relative higher buried volume %V_bur_ than **Ti1** (55.0% vs. 54.2%), implying the more sterically crowded environment around the titanium atom, which prevented ethylene monomer from accessing to the metal center and thus eventually resulted in inferior catalytic activities. Additionally, because of the steric congested reason that is able to suppress chain transfer reaction, polyethylene products obtained from **Ti3**/DMAO revealed much higher molecular weight than **Ti1** and **Ti2** mediated systems (M_w_ = 26.2 × 10^4^ g/mol (**Ti1**), 21.8 × 10^4^ g/mol (**Ti2**), 79.4 × 10^4^ g/mol (**Ti3**)).

Another thing worthy of note is that, for half-titanocenes bearing 2,6-unsubstituted aryloxides, such as CpTiCl_2_(O(4-^t^BuPh) and CpTiCl_2_(O(4-MePh), very low catalytic activities generally resulted in olefin polymerization [48]. Such catalytic inefficiencies were probably due to the lack of bulkier ortho-substituents that could force a more open Ti-O-C_ipso_ bond angle, which finally led to less O→Ti donation into Ti atom and therefore destabilized the active species. For the present complexes **Ti1** and **Ti3**; however, although their unstacked structures exhibited similar buried volume %V_bur_ to CpTiCl_2_(O(4-^t^BuPh) (Figure 6, 52.0%, 52.2% vs. 50.7%), appreciable catalytic efficiencies were eventually afforded. These satisfying results were also presumably due to the big π systems caused by π-π stacking interactions, which were able to enhance the electron donation to the metal center and therefore gave a more stable catalytic active species.

The most active precatalyst was concluded to be **Ti4** bearing pentamethylcyclopentadienyl (Cp*) and 1-pyrenoxide ligands, in which a catalytic activity of 5.14 × 10^6^ g PE•mol^−1^ (Ti)•h^−1^ was demonstrated. This was consistent with Nomura’s results that the more electron donating Cp* was able to stabilize the active species, and thus led to higher activity [31].

π-π stacking conformations are very sensitive to high temperatures. Generally, the stacked dimer structure tends to be dissociated upon increasing the temperature. Therefore, in order to better elucidate the influence of the π system on catalytic performances, ethylene polymerization at different temperatures were carried out by using **Ti3** and **Ti4** at precatalysts. As the data shown in Table 1 and Figure 7, upon increasing the temperature from 20 °C to 70 °C, both of **Ti3** and **Ti4** revealed a first increasing and then decreasing trend, with 50 °C as the optimized temperature. Such increasing polymerization activities from 20 °C to 50 °C were probably due to the dissociation of π-π stacking structures into unstacked active species, which allowed more monomers to access to the metal center, as revealed from the decreased buried volumes %V_bur_ when comparing the stacked and unstacked complexes (55.0% vs. 52.2% for **Ti3**). Further increasing polymerization temperature to 70 °C witnessed obviously decreased activities for both two complexes, which were presumably due to the decomposition of the active species at very high temperatures. Moreover, elevating polymerization temperature also posed big influence on the molecular weights of the resultant polyethylenes. For **Ti3** and **Ti4** mediated polymerizations, a first increasing and then decreasing trend was also observed for the resultant polymer products when increasing the temperature from 20 °C to 70 °C.

Inspired by the good catalytic efficiencies of **Ti3** and **Ti4** towards ethylene polymerization, their performances for copolymerization of ethylene and 1-hexene was also evaluated. As the data summarized in Table 2, moderate to high catalytic activities in the range of 1.08–10.26 × 10^6^ g polymer•mol^−1^ (Ti)•h^−1^ were obtained. Compared to ethylene homopolymerizations, **Ti3** revealed comparable copolymerization activities when 0.32 mol/L comonomer was introduced, further increasing the 1-hexene concentration to 0.48 mol/L resulted in a decreased catalytic activity to 1.08 × 10^6^ g polymer•mol^−1^ (Ti)•h^−1^. In contrast, **Ti4** revealed distinctly different copolymerization behaviors. With an increasing 1-hexene concentration from zero to 0.48 mol/L, **Ti4** demonstrated monotonously increased catalytic activities from 8.19 × 10^6^ g polymer•mol^−1^ (Ti)•h^−1^ to 10.26 × 10^6^ g polymer•mol^−1^ (Ti)•h^−1^. When further increasing 1-hexene concentration to 0.70 mol/L, its activity was hardly changed. The much-improved catalytic activities with increasing 1-hexene concentrations for **Ti4** was probably ascribed to the comonomer effect, which allowed more monomers to access to the active species and thus more enchainment possibilities. Determined by ^13^C NMR (Figure 8), the 1-hexene incorporation levels in the resultant copolymers were in the range of 8.1–15.6%, and comonomer sequence analysis for copolymer samples can be found in Table 3. Determined by DSC analysis, the T_m_ values of the copolymers obtained from **Ti4** decreased gradually from 131 °C to 62 °C, and the DSC curves changed from a sharp peak to broad melting range, indicating the randomly incorporated 1-hexene commoners.

Considering the positive influence of π,π-stacking on ethylene polymerizations, when comparing with the half-titanocenes counterparts bearing 2,6-unsubstituted aryloxides, we are trying to explore whether externally added π-conjugated small molecules, which will also form π,π-stacking interaction with the fused-aryloxide moieties in the present titanocene complexes, would also enhance the catalytic performances. Based on this, **Ti1**–**Ti3** catalyzed ethylene homopolymerizations were carried out in the presence of 1.0 equiv. of pyrene. As shown by the data in Table 4, obviously enhanced catalytic activities were observed for all the three systems, giving increased values from 4980 to 5190 g PE•mol^−1^ (Ti)•h^−1^ for **Ti1**, from 5070 to 5760 g PE•mol^−1^ (Ti)•h^−1^ for **Ti2**, from 1680 to 2610 g PE•mol^−1^ (Ti)•h^−1^ for **Ti3**, respectively. Moreover, molecular weights of the resultant polyethylenes were also much higher than pyrene-free systems (26.2 × 10^4^ vs. 37.5 × 10^4^ for **Ti1**, 21.8 × 10^4^ vs. 49.3 × 10^4^ for **Ti2**, 79.4 × 10^4^ vs. 111.7 × 10^4^ for **Ti3**), indicating the formed active species therein were more stable and therefore long-lived. These results could be explained by the assumption that the original π,π-stacked dimer of complex **Ti3** would be dissociated in the presence of pyrene molecules and then restack with pyrene to form a more active and stable active species (Figure 3). Such a speculation could be established after comparing optimized structures of π,π-stacked dimer of complex **Ti3** and **Ti3**-pyrene shown in Figure 3 (bottom), in which the latter one revealed a relative stronger π,π-stacking interaction than the former one, as evaluated from the distances between two almost parallel planes (3.267 Å vs. 3.318 Å), implying that **Ti3** revealed a bigger tendency to stack with pyrene molecule rather than itself. Such an observation made sense when considering the bigger electron density in pyrene than the pyrenoxide group that was connected to an electrophilic titanium metal center. Because of the same reason, the C_ipso_-O-Ti bond angle of **Ti3**-pyrene was slightly higher than that in **Ti3** dimer (156.9^o^ vs. 156.5^o^). Additionally, the reformation process of π,π-stacking interaction between **Ti3** and pyrene could be also monitored by in situ NMR and UV/Vis studies, which had been reported in other related complexes [15,17,63]. In the UV/Vis experiment, the concentration of **Ti3** was gradually increased from zero to 24 × 10^−7^ M while keeping the concentration of pyrene unchanged (5 × 10^−6^ M). It was found that the intensity of the absorbance of pyrene was gradually enhanced (Figure 9), indicating the formation of strong binding, i.e., π,π-stacking interaction, between **Ti3** and pyrene molecules. This interaction could be also evidenced from the NMR study, which was carried out by gradually adding pyrene (0–3.9 equiv.) to a C_6_D_6_ solution of **Ti3** (17.8 mM). As the spectra shown in Figure 10, two characteristic proton resonance peaks at ca. 8.65 ppm and ca. 7.35 ppm, which were assigned to the 6- and 2- substituted protons on the pyrenoxide group, respectively, witnessed a clear upshift to high field when gradually adding more pyrene molecules. This was due to the formation of a stacking interaction between **Ti3** and pyrene, which caused a bigger shielding effect due to it having a bigger electron density than the pyrenoxide moiety.

## 4. Conclusions

In summary, we have prepared a series of half-titanocene complexes containing fused-aryloxide ligands. Due to the presence of big π-systems therein, such complexes could form π,π-stacking interactions to give dimer structures, and such interactions could be clearly observed from single crystal X-ray spectroscopy analysis. Because of these π,π-stacking interactions, the present half-titanocenes revealed good catalytic activities to ethylene homopolymerizations and copolymerization with 1-hexenes, which confirmed the positive influence of π,π-stacking interaction on enhancing the catalytic performances when comparing with other half-titanocenes bearing 2,6-unsubstituted aryloxide moieties. Moreover, the overall catalytic behaviors of these complexes can be regulated by adding external pyrene additives. Through formation of a stronger π,π-stacking between the complexes and pyrene additives, the catalytic efficiencies as well as the molecular weight of the obtained polymers could be further enhanced.

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
