# Peer review of "Synthesis of Half-Titanocene Complexes Containing π,π-Stacked Aryloxide Ligands, and Their Use as Catalysts for Ethylene (Co)polymerizations"

_polymers, 2022, doi:10.3390/polym14071427_

Round 1

Reviewer 1 Report

  • Figures need to be enlarged.
  • Some linguistics errors need to be revised.
  •  

Author Response

Thank you very much for your comments. We carefully read your comments, and our revisions and replies are as follows:

  1. Figures need to be enlarged.

All the figures had been enlarged according to the reviewer’s comment.

  1. Some linguistics errors need to be revised.

Thanks for your valuable comments. We have went through the text and corrected the linguistic errors accordingly.

Reviewer 2 Report

In this manuscript, the authors described several half-titanocene complexes containing π,π-stacked ligands, and their use as catalysts for olefin polymerizations. This report might be suitable for publication given if the following concerns could be addressed:

  1. For ethylene polymerization reactions, the authors reasoned that the inferior performance of Ti3 compared to that of Ti1 is because of the more sterically crowded environment around the titanium atom for Ti3. However, Ti2, which is sterically more crowded compared to Ti1 actually performed better than Ti1 for PE synthesis. It seems like there are reasons other than simply steric that causes this trend (catalytically Ti2>Ti1>Ti3 while sterically Ti3>Ti2>Ti1). Additionally, the authors did not perform the Steric crowding maps calculations for Ti2 (only Ti1 and Ti3 were calculated in Figure 5). The authors should perform this calculation for Ti2 and add to Figure 5. The difference between the %Vbur of Ti1 compared to that of Ti3 is small (55.0% vs 54.2%), so steric should not be the only reason causing such a significant difference in terms of catalytic performances towards PE synthesis.
  2. Considering the good catalytic activity of Ti1 and Ti2, high temperature experiments should also be performed by using these two catalysts to draw better conclusion regarding the catalytic performances of these Ti complexes vs. the pi-pi stacking. Has similar behavior (increase of catalytic activity from 20 to 50 C) previously been observed in the literature for non-pi-stack containing Ti complexes for PE synthesis? If so, then the conclusion regarding the disruption of pi-stacking that is causing this behavior needs to be reevaluated and more control experiments will be needed. 
  3. Please provide units for Mw of the polymers (both in the tables and in the text).

Author Response

Reply to the comments by Reviewer 2

Thank you very much for your comments. We carefully read your comments, and our revisions and replies are as follows:

  1. For ethylene polymerization reactions, the authors reasoned that the inferior performance of Ti3 compared to that of Ti1 is because of the more sterically crowded environment around the titanium atom for Ti3. However, Ti2, which is sterically more crowded compared to Ti1 actually performed better than Ti1 for PE synthesis. It seems like there are reasons other than simply steric that causes this trend (catalytically Ti2>Ti1>Ti3 while sterically Ti3>Ti2>Ti1). Additionally, the authors did not perform the Steric crowding maps calculations for Ti2 (only Ti1 and Ti3 were calculated in Figure 5). The authors should perform this calculation for Ti2 and add to Figure 5. The difference between the %Vbur of Ti1 compared to that of Ti3 is small (55.0% vs 54.2%), so steric should not be the only reason causing such a significant difference in terms of catalytic performances towards PE synthesis.

Thanks for your comments, steric crowding map calculations of complex Ti2 cannot be carried out because its single crystal structure were not isolated successfully.

Regarding the catalytic activities of Ti1-Ti3, they are mainly influenced by two factors: (1) steric effects of the fused-aryloxide ligand, (2) the Cipso-O-Ti bond angles, in which a larger angle implies O→Ti π donations into titanium and therefore higher reactivities (Dalton Trans. 40 (2011) 7666-7682; Polymer 100 (2016) 188-193.) It seems that an appropriate combination of these two factors will eventually give rise to optimist catalytic activities. Because for Ti3, although it has much bigger Cipso-O-Ti bond angle of 158.26(18)o, its most sterically congested environments at the metal center, caused by the π-π stacking of pyrenoxide group, renders it poorest catalytic activities eventually. Whereas for complex Ti1, its least congested active species results in relative higher catalytic activities, regardless of the much smaller Cipso-O-Ti bond angle of 158.26(18)o. Therefore, we have reasons to postulate that, complex Ti2, that reveals medium value of Cipso-O-Ti bond angle and suitable steric congestion finally gives rise to highest catalytic activities.

  1. Considering the good catalytic activity of Ti1 and Ti2, high temperature experiments should also be performed by using these two catalysts to draw better conclusion regarding the catalytic performances of these Ti complexes vs. the pi-pi stacking. Has similar behavior (increase of catalytic activity from 20 to 50 C) previously been observed in the literature for non-pi-stack containing Ti complexes for PE synthesis? If so, then the conclusion regarding the disruption of pi-stacking that is causing this behavior needs to be reevaluated and more control experiments will be needed. 

Thanks for your valuable comments. Only complex Ti3 and Ti4 that bears strongest π-π stacking interactions was evaluated at different polymerization temperatures, because such a stronger interaction was more sensitive to polymerization temperature. No variable temperature experiments had been carried out for complex Ti1 and Ti2 because (1) Ti1 and Ti2 demonstrated much weaker π-π stacking interactions as concluded from single crystal structure analysis; (2) previous reports had concluded that half-titanocene metallocene complexes bearing unstacked phenoxide moieties revealed much poorer activities along increasing polymerization temperature. (Journal of Molecular Catalysis A: Chemical 254 (2006) 197–205.)

  1. Please provide units for Mw of the polymers (both in the tables and in the text).

All the units of Mw had been added into the manuscript according to the reviewer’s comment.

Round 2

Reviewer 2 Report

The authors have addressed all my previous comments and concerns. The manuscript is ready to be published.